**Data Availability Statement:** Data is available at https://doi.org/10.3886/E119470V2.

**Funding:** Publishing current paper was financed through Entrepreneurial Education and Professional Counseling for Social and Human Sciences PhD and Postdoctoral Researchers to

# Sociological dimensions of marital satisfaction in Romania

**Delia Nadolu**[1]*, **Remus Runcan**[2], **Aurel Bahnaru**[3]

**1** Department of Sociology, West University of Timisoara, Timisoara, Romania, **2** Faculty of Sociology and Psychology, West University of Timisoara, Timisoara, Romania, **3** West University of Timisoara, Timisoara, Romania

* delia.nadolu@e-uvt.ro

## Abstract

We are living nowadays in a social paradigm characterized by a high degree of fluidity. From professional career to leisure, from family patterns to neighborhood relationships, from cultural consumption to domestic technology, almost all the components of social reality have changed during recent decades. A given couple's experience is not insulated from these dynamics, or at least from the pressure that new trends constantly put on it. How can functional relationships be preserved in a continuously changing world? What possibilities are there for couples to sustain viable relationships in the face of all the waves of change, involving as they do new content, new rules, and, in many cases, new values? This paper sets out to analyze how the main factors related to marital life interact and what their impact is on individual satisfaction in the dyadic experience. To this end we planned and applied a sociological survey to a national sample (N = 455 participants, error limit 4.7) using a questionnaire focusing on an evaluation of dyadic life experience that included the Dyadic Adjustment Scale (DAS). The major finding is that more liberal sexual attitudes and people's high view of the importance of money are the strongest predictors of a low-quality dyadic experience. The patterns observed also raise the possibility that positive perception of the parental model may serve to compensate for a couple's relatively shorter period of marital experience.

## Introduction

Marital satisfaction and its correlates have been investigated almost exclusively in Western countries [1]. Marital relationships are strongly conditioned by culturally determined norms, customs, and expectations.

Marital satisfaction refers to an individual's global evaluation of the marital relationship [2]. Durodoye (1997) defined marital satisfaction as an individual's subjective evaluation of the specific components within his/her marital relationship [3], while Fatehizadeh & Ahmadi found that marital satisfaction plays a major role in the stability of a marriage [4]. Garcia (1999) believes that satisfaction is to be considered at three levels: general satisfaction with life, satisfaction with family relationships, and satisfaction with one's spouse [5].

ensure knowledge transfer Project, financed through Human Capital Programme (ATRiUM, POCU380/6/13/123343).

**Competing interests:** The authors have declared that no competing interests exist.

Attitudes towards marital relationships have been examined in three ways in the literature: expectations of what married life will be like, global positive or negative attitudes towards marriage, and intent to marry [6]. Marital relationships have also been scrutinized from the perspective of the relationship between stability and marital satisfaction [7, 8], of the relationship between equity and marital satisfaction [9, 10], and from the perspective of the quality of premarital relationships, of the quality of marriage, of the way the members of a couple relate to each other, of couples' personality types, and of the way they deal with problems in their marriage [4]. A satisfactory relationship is the most important and complex aspect of intimate relationships.

According to Daneshpour, strong marriages are predicted by such factors as closeness, communication, conflict resolution, family and friends, financial management, flexibility, leisure activities, personality issues, sexual relationship, and spiritual beliefs. Park claims that the factors that ensure marriage sustainability are commitment, communication, emotional support, fidelity, finances, fulfilment, having children, respect, romance, sexual intimacy, shared values, and trust [11].

Fig 1 illustrates some key socials measure of marriage and divorce trends in Romania since the fall of communism. Instructive contextual comparison with other European countries can be made with the data shown in Fig 2. It is noteworthy that the rate of marriage in Romania has remained high in comparison with other middle and eastern European countries. Cultural, religious and economic factors are all involved here, Romania being a social conservative country under the influence of a preponderant tradition of Orthodox Christianity. It is also reasonable to speculate that widening educational opportunities and longer training and career development trajectories have influenced the steadily rising trends in average marriage age in both sexes.

## Materials and methods

In this paper, we analyze only five of the factors that contribute to marriage satisfaction: parental pattern of the family of origin, spirituality, sexuality, material resources, and couple experience (duration and children).

### 2.1. Parental pattern of the family of origin

A family of origin is "a family into which a person is born" [12]. Marital instability later in life has been attributed to family traumas experienced in childhood (abandonment by biological

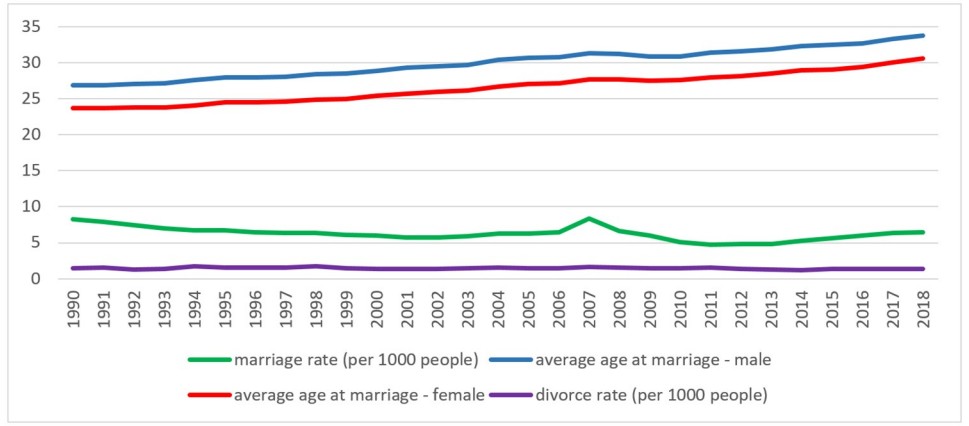

**Fig 1. Trends in Romanian marriage and divorce rate data since 1990 (Romanian National Statistics Institute).**

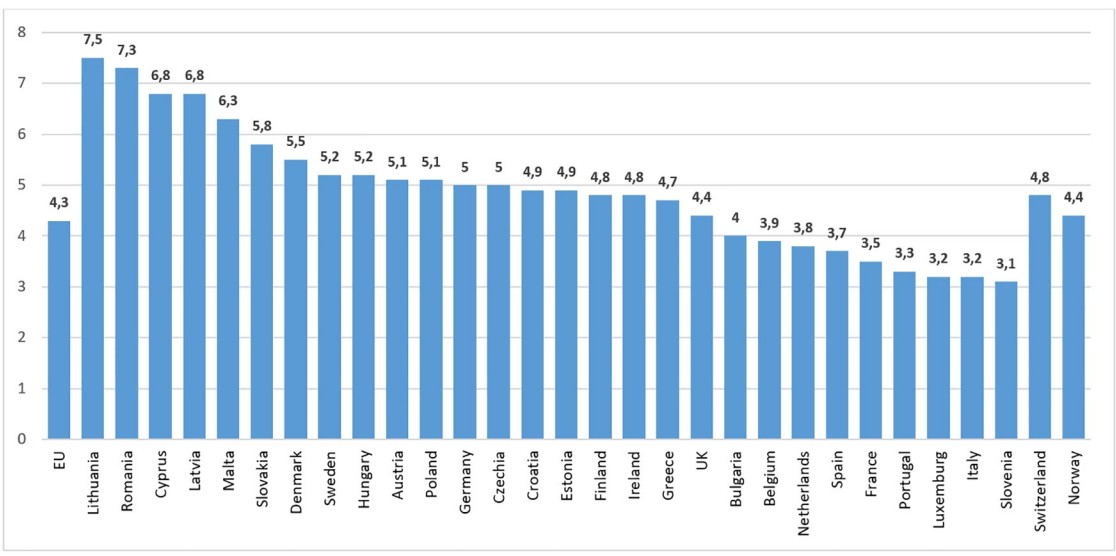

**Fig 2. Romanian statistics in a wider European context—Marriage rate per 1000 people (Eurostat).**

parents, alcoholic parents, death of a parent, death of a sibling, mental illness of a parent, physical abuse, sexual abuse, teenage pregnancy before marriage) [13, 14].

## 2.2. Spirituality

Almost a decade ago, researchers recognized that two key issues had previously been insufficiently studied: the impact of infidelity, and the role of religion in marriage [15]. It is important to note that honesty, loyalty, morality, and religiosity are variables associated with fidelity [16], one of the most important prerequisites of a sustainable marriage. Previously, some researchers had even stated that fidelity in marriage is similar to religious faith [17].

Faith in God is among the core factors positively associated with marriage stability, together with commitment, cooperation, effort and perseverance, friendship, honesty, independence, mutual understanding, patience, problem-solving ability, self-sacrifice, preserving a distance from compassionate and non-compassionate sympathy or interference/intrusion of others in the couple's life, the presence of children, and trust [4]. Likewise, compatible religious beliefs are among the factors predicting marital satisfaction. Other key factors include enjoyment of shared fun/humor, love, loyalty/fidelity, mutual give and take, mutual respect, mutual support, mutual trust, shared interests, and having similar philosophies of life [18]. Moreover, for couples who share similar religious beliefs, spiritual practices can be shared practices that sustain and improve the marital relationship [19, 20]. Couples who share a spiritual orientation are more likely to pray together and attend church together, which can be regarded as two indicators of a positive marital relationship [4].

In a large scale metastudy associations have been reported between religious involvement and spirituality and the reinforcement of marital relationships; these have been interpreted as arising from the meaning and structure such involvement brings to couple relationships [21, 22]. These authors also acknowledge that there are potential mechanisms connecting religion and spirituality to marital relationship quality. Couples reported four predominant social contexts that give sacred meanings to their marriage: communion, first-rite rituals, prayer, and worship services and sermons [15].

Religious teachings can strengthen marital relationships because they emphasize forgiveness, care for others, and the restraint of anger, all of which foster positive individual attitudes [21]. A study of marital fidelity based on analysis of in-depth interviews with highly religious couples identified four important ways in which such couples understood the relevance of their religious involvement [23]:

Religion strengthens couples' moral values, which promotes fidelity;

Religious belief and practice sanctify marriage and thereby improve marital quality, which indirectly promotes fidelity;

Religious involvement improves spouses' relationship with God, which encourages them to avoid infidelity (something they believe would displease God);

Religious vows and involvement strengthen commitment to marital fidelity.

## 2.3. Sexuality

It is worth mentioning that traditional couples emphasize the importance of the sexual relationship in strengthening their marital relationships [4]. The effects of sexual intimacy on marital relationships are essential, as is the causal sequence whereby sexual satisfaction influences marital relationships in a similar way for both men and women, even though sex may have different meanings for men and women in their relationship [24, 25].

## 2.4. Material resources

Proper financial management is a good predictor of strong marriages [4]. Wong & Goodwin [4] demonstrated that in Britain, Hong Kong, and China, a stable relationship with the spouse, partnership, spousal support, and stable family finances are important factors contributing to marital satisfaction. Alongside emotional support (caring, staying optimistic), intimacy, loyalty (trustfulness, staying true and respect), similarity (common interests and goals), a sense of (economic) security is recognized by couples as a crucial factor in making a relationship sustainable and satisfactory. Economic security is seen as especially important for married couples, whereas emotional support such as caring and staying optimistic is viewed as beneficial and more appealing to young, dating couples [26, 27].

## 2.5. Couple experience (duration and children)

Good inner personality traits, physical beauty, and unique romantic experiences seem to be the common factors contributing to attraction and sustainable relationship [26]. Successful / sustainable marriage is defined by [4]: adjustment (whether spouses have acquired the life skills needed for marriage); commitment (reflecting a person's desire to stay in a relationship, which is affected by someone's attraction to their partner, the relationship, and the couple identity); duration (the number of years that a marriage remains intact); satisfaction (the extent that each spouse feels internal joy, contentment, and love in their relationship); and stability (whether a marriage remains intact or is dissolved).

Having children or the presence of children is one of the factors most necessary for keeping a marriage stable, alongside dedication, faith in God, forbearance, love, and patience [4].

To summarize, the factors conducive to happy, satisfying, and successful marriages are [4]: consulting with each other, cooperating with each other in children's upbringing, expressing love to each other, sharing common beliefs, the couple solving their own problems, perceiving the relationship as intimate, trusting each other, and being committed.

## Methodology

The central research question of the study reported here was: What are the main factors that have a significant impact on marital satisfaction in contemporary Romanian society? With this objective in mind we designed a two-part sociological questionnaire about marital satisfaction: 26 questions related to the evaluation of a number of factors directly associated with the quality of marital life, and, separately, the Dyadic Adjustment Scale (DAS) devised by Graham Spanier in 1976.

The sociological survey field research was conducted between August and December 2018 at national level, on a representative sample (N = 455, with an error limit of 4.7) balanced and stratified for age, gender and urban/rural environment, randomized for location and with a fair representation of the country's historic regions. Respondents were individuals rather than couples, and data analysis took appropriate account of their current or previous relationship status when analyzing these individuals' reported couple relationship satisfaction levels. The questionnaire was anonymous and administered on paper (Annex 1). All participants were informed about the objective of the study and consented to completing the questionnaire.

The data was input to the SPSS package and statistical analysis was carried out in accordance with accepted methodology.

## Ethics statements

Permission to carry out this research project, under the title Sociological Dimensions of Marital Satisfaction in Romania, was sought, on behalf of the research team, by Remus Runcan from the Ethics and Standards Supervisory Board of the University of the West, Timişoara (application request number 1952/0-1/16.01.2020, RCE 2020–7). Details were also furnished of the method proposed, the research hypothesis, the procedure for administration of the printed questionnaire, sampling and proposed statistical processing of the results (using anonymized data and the SPSS program suite). A copy of the questionnaire was provided. The Board certified that these met the criteria for a research project and that they were in conformity with required standards for the storage and processing of personal data. Formal permission to proceed was granted under registration number 2633/0-1/20.01.2020, RCE2020-16.

## Results

In the interpretation and discussion of our results we have kept in mind that when reference is made to the life of the couple or to the quality of the couple relationship this is to be understood within the limits of our research methodology, which involved responses from individuals concerning their perceptions of the relationships they were, or had been, in.

The relationships between the following five variables and the quality of dyadic life of a couple were analyzed:

first variable, parents = the parental model, found by adding the scores for responses to two questions: perceived / reported relationship between the respondent and their mother/ father, and the reported relationship between the respondent's parents (both answers being scored on a 5-point Likert scale). These two components show a strong positive correlation (r = 0.587, p<0.001) and reflect the influence of the parental model on dyadic life.

the second variable, God = spirituality, is based on a single question with a YES/NO answer (Do you consider that belief in God can reduce the risk of ethical deterioration in marital life?).

the third variable, sex = attitude toward sexuality, was realized by combining the scores in response to three questions (all with answers on a 5-point Likert scale): acceptance of pre-marital cohabitation, acceptance of living together, and acceptance of sexual relationships before marriage. The association of the responses to these three questions is admittedly quite weak (KMO = 0.55) but it was decided to continue to use them all in the composite variable because personal attitudes toward sexual life (conservative vs permissive) are an important dimension of dyadic existence. We acknowledge that these indices of sexual permissiveness are all retrospective (referring to attitudes towards pre-marital sexual behaviors) and that this research study made no attempt to measure attitudes, whether conservative or permissive, towards such issues as marital faithfulness and sexual exclusivity. Such a study, though possible, would need careful design and validation in a still socially conservative country and would not, we feel, have added materially to a preliminary analysis of the influence of this attitudinal dimension.

the fourth variable, money = the importance of money, relates to the response (scored on a 5-point Likert scale answer) to a single question (Do you consider that having a reasonable standard of living can maintain a couple's relationship?).

the fifth variable, experience = the effect of prior experience on the dyadic experience combines two elements in the data: the duration of the most significant, perhaps previous, relationship and the presence or absence of children. When computing this parameter, we decided to apply a multiplying coefficient of 1.5 to the relationship duration if the respondent had at least one child. We justify this on the grounds that the presence of one or more children gives a more complex dyadic experience in comparison with relationship situations which do not involve children.

The graphs in Fig 3 show the percentage distribution of data for each of these variables and for the DAS score.

There are strongly significant negative correlations between belief in God and permissive attitudes to sex (-0.377) and between belief in God and expressed attitude to the importance of money (r = -0.196, p<0.001). Also, there is a strongly significant positive correlation between these attitudes to sex and to money (r = 0.292, p < .001). All these factors are in some way correlated with the general DAS score (Table 1):

This table shows that high levels of dyadic satisfaction are positively associated with the respondents' favorable appreciation of their parental model and also with belief in God. Conversely, there are negative associations between the aforementioned attitudes towards sex and money and favorably reported dyadic experience. In other words, a favorable parental model and a belief in God can be considered predictors for a positive climate within a couple, whereas liberal attitudes to sex and a focus on money show a negative relationship. Paradoxically, a more lengthy dyadic experience seems to contribute to a decrease in couple satisfaction. We have needed to address the problem of missing data arising from partial responses, but given the limited residual sample sizes have not used an imputation technique, preferring to work directly from the admittedly more restricted data available. It should be explained that the differences of N values arise as a consequence of the composite structure of each dimension and the need for full responses to all the relevant components. For instance, when calculating the correlation of DAS score with experience (where N = 183) only responses that included answers to both variables were processed. Furthermore, the DAS score was calculated only for subjects who had responded to all 32 items. The DAS items with the highest level of non-answers were I30 (Not showing love) (N = 336) and I29 (Being too tired for sex) (N = 347). Since in 102 cases responses to both questions were not given, our inference is that there was a

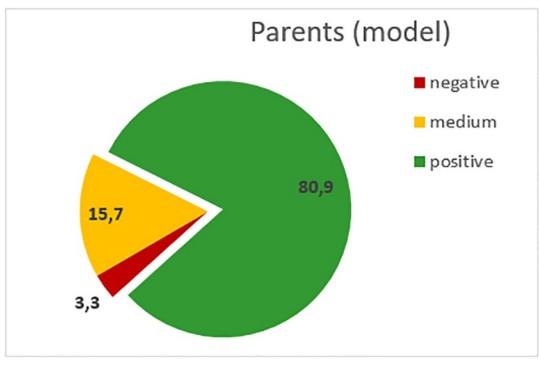
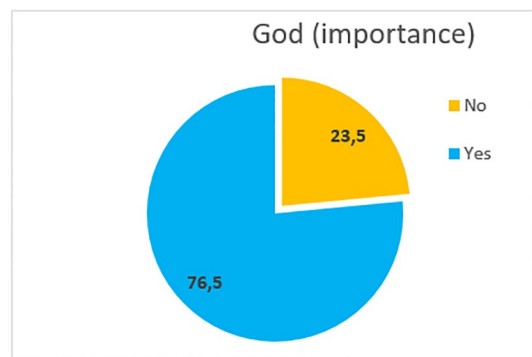
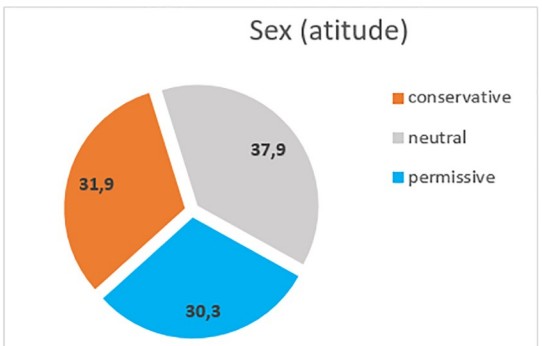
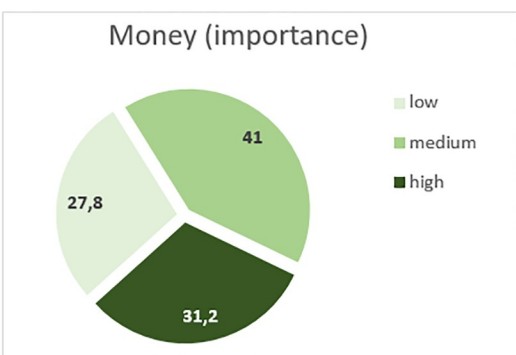
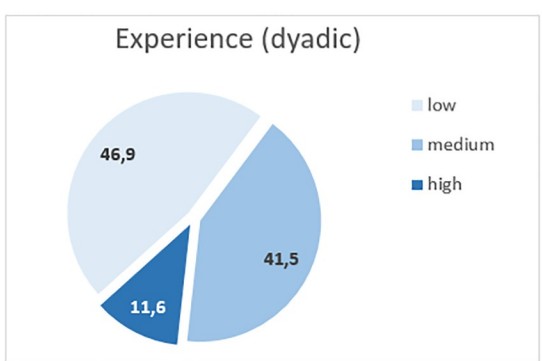
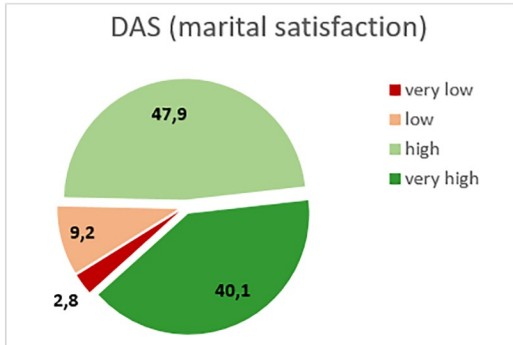

**Fig 3. Descriptive statistics of computed variables and DAS score.**

double effect: the nature of the items (too personal), combined with their position at the end of the questionnaire. For the item related to prior dyadic experience the quite high number (86) of missing responses arose from the fact that only subjects with at least six months' experience were taken account of.

**Table 1. Correlations between DAS score and computed variables.**

| | | parents | God | sex | money | experience |
|---|---|---|---|---|---|---|
| DAS | Pearson correlation | 0.307** | 0.201** | -0.239** | -0.304** | -0.161* |
| | Sig. (2-tailed) | 0.000 | 0.002 | 0.000 | 0.000 | 0.030 |
| | N | 281 | 233 | 244 | 282 | 183 |

** = Correlation is significant at the 0.01 level (2-tailed).

* = Correlation is significant at the 0.05 level (2-tailed).

**Table 2. Component matrix.**

| | Component | |
| --- | --- | --- |
| | **F1** | **F2** |
| sex | 0.809 | 0.019 |
| God | -0.758 | 0.179 |
| money | 0.637 | 0.416 |
| parents | 0.021 | -0.676 |
| experience | -0.202 | 0.647 |

**Table 3. Correlations between DAS score and derived complex factors.**

| | | F1 | F2 |
| --- | --- | --- | --- |
| DAS | Pearson correlation | -0.361** | 0.483** |
| | Significance (2-tailed) | 0.001 | 0.000 |
| | N | 83 | 83 |

A fuller analysis of the sex factor (attitude toward sexuality) and its correlation with the relevant components of the DAS did not bring to light any significant associations: sex factor with I6 Sexual relationship r = .064, sex factor with I7 Conventionality r = .0036, sex factor with I23 Kissing the mate r = .062.

Analysis of these factors in relation to several other factual variables also shows the following:

there are no significant differences between genders for any of the five factors—chi square tests not significant;

there is only one significant difference related to age, which is understandably–since older people will have more experience—with the experience factor (it is actually almost a collinearity)–Anova test F = 122.830, df = 2, sig< 0.001

there is a significant correlation (albeit with a low value) between the sex factor and the duration of the relationship r = .168, sig < 0.01, N = 312

A factor analysis carried out for these five variables shows a quite strong association among them (KMO = 0.594) and generates two main factors (F1 and F2 in Table 2) that account for 55% of the total variance (Extraction Method: Principal Component Analysis. a. 2 components extracted.):

The first complex factor can be summarized as describing a pragmatic non-spiritual approach (F1): an interest in money, an open-minded mentality regarding sexual issues and a disinterest in the spiritual dimension. The second complex factor (F2) is less easily characterizable, although both component variables are more external; it involves a conservative approach based on the available parental model. Both factors show significant correlation with DAS–the ** mark indicating that the correlation is significant at the 0.01 level (2-tailed)–see Table 3. The smaller N (83) in this analysis arises for the same reasons as those explained previously in relation to Table 1.

## Discussion

On the basis of our research results, the practical sustainability of the contemporary marital model can be seen as involving two competing approaches: (1) a pragmatic non-spiritual

approach and (2) a conservative approach based on the available parental model. The first of these focuses on the valorization of sex and money, perceived as resources for the quality of the relationship. However, these two components show negative correlations with the satisfaction scores declared for the experience of the couples' lives. The second may reflect a valuing of the parental model to compensate for more limited duration of dyadic experience (or, mutatis mutandis, a lesser focus on the parental model once there is sufficient dyadic experience). But, whereas the parental factor has a positive correlation with the DAS score, the experience variable shows a negative association with it. We thus observe a complex effect from these two variables; a positive reported parental model is positively associated with perceived dyadic satisfaction, while duration of couple experience can have either type of consequence: positive or negative. We might consider that during any long period living together a wide variety of situations are encountered, some involving the resolution of serious problems and misunderstandings. Thus, not even experience, as we have defined it, can be used as a direct predictor of dyadic satisfaction; this variable combines with the effects of the parental model, and we may suggest that people lacking experience are more likely to report having had positive parental models, and we may hypothesise that this evaluation contributes to their desired sense of marital satisfaction.

The other factor is easier to interpret. It brings together three variables–a permissive attitude towards premarital sex, a prioritization of money, and the index for belief in God (which shows a negative degree of association). So, we can see that people who have a liberal attitude towards sex and regard money as an important factor for the maintenance of family life tend to place a lower value on the religious dimension. The effects of these three variables on marital life satisfaction differ, with liberal sexual attitudes and a preoccupation with money having a negative correlation, while belief in God has a positive one.

## Conclusion

Satisfaction in marital life is a complex phenomenon that depends on a number of variables. Where there is lack of dyadic experience, as in the case of young couples, a positive appreciation of the parental model is strongly correlated with high-satisfaction marital life. An open attitude toward sexual life (although not unfaithful behavior) and a belief in the importance of money are indicators of low-satisfaction marital life. The religious dimension (belief in God) is also a predictor of high marital satisfaction.

At first glance, this may appear to reflect simply a traditional and interculturally dated mode of life, although it is in fact not far from the Romanian marital model [28]. However, deeper reflection might lead us to see that an acceptance of sexual relationships before marriage, and of living together without getting married (two very common contemporary patterns), both clearly indicate an implied assumption that one relationship can be ended and another started if things are not going well. This is increasingly standard behavior nowadays, when the quality of each individual's life is seen as more important than the preservation of socially desirable patterns. In other words, if a serious problem arises, quitting a relationship in difficulty is viewed as a more viable option than trying to maintain the marriage at all costs. In this context, the integrity of the marital relationship is no longer perceived as a social good conducive to sustainability; rather, such relationships involve very fragile and fluid configurations that may change at any point during a couple's life together.

## Supporting information

**S1 File. English questionnaire.**
(DOCX)

**S2 File. Romanian questionnaire.**
(DOCX)

## Acknowledgments

Special thanks to Stuart and Dorothy Elford for proof reading in English, as well as for their constructive feedback regarding specific aspects of the research methodology.

## Author Contributions

**Conceptualization:** Remus Runcan, Aurel Bahnaru.

**Methodology:** Delia Nadolu, Aurel Bahnaru.

**Resources:** Remus Runcan.

**Validation:** Delia Nadolu, Remus Runcan, Aurel Bahnaru.

**Writing – original draft:** Delia Nadolu, Remus Runcan.

**Writing – review & editing:** Delia Nadolu, Remus Runcan, Aurel Bahnaru.

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
