## [Decision Letter · Decision Letter 0]

4 May 2020

PONE-D-20-02203

SOCIOLOGICAL DIMENSIONS OF MARITAL SATISFACTION IN ROMANIA

PLOS ONE

Dear Dr. Delia,

Thank you for submitting your manuscript to PLOS ONE. After careful consideration, we feel that it has merit but does not fully meet PLOS ONE’s publication criteria as it currently stands. Therefore, we invite you to submit a revised version of the manuscript that addresses the points raised during the review process.

The revised version should address all the comments in the reports.

We would appreciate receiving your revised manuscript by Jun 18 2020 11:59PM. To enhance the reproducibility of your results, we recommend that if applicable you deposit your laboratory protocols in protocols.io, where a protocol can be assigned its own identifier (DOI) such that it can be cited independently in the future. For instructions see: http://journals.plos.org/plosone/s/submission-guidelines#loc-laboratory-protocols

We look forward to receiving your revised manuscript.

Kind regards,

Petri Böckerman

Academic Editor

PLOS ONE

Journal Requirements:

3. We note you have included a table to which you do not refer in the text of your manuscript. Please ensure that you refer to Tables 2 and 3 in your text; if accepted, production will need this reference to link the reader to the Tables.

Additional Editor Comments (if provided):

Reviewers' comments:

Reviewer's Responses to Questions

**Comments to the Author**

1. Is the manuscript technically sound, and do the data support the conclusions?

Reviewer #1: Yes

Reviewer #2: Yes

2. Has the statistical analysis been performed appropriately and rigorously? 

Reviewer #1: Yes

Reviewer #2: No

3. Have the authors made all data underlying the findings in their manuscript fully available?

Reviewer #1: Yes

Reviewer #2: No

4. Is the manuscript presented in an intelligible fashion and written in standard English?

Reviewer #1: Yes

Reviewer #2: Yes

5. Review Comments to the Author

Reviewer #1: Being from the US, I would have liked more information on marriage in general in Romania. This would have been beneficial to having a better comprehension of the findings in the study.

Reviewer #2: There are some things that should be done to make the data analysis clearer and more complete.

1. Sample size issues. Although there were 455 persons who responded to the survey as indicated in your methodology section, the Ns reported in the tables were substantially less. In Table 1 the highest N was 282, and in Table 3 it was only 83. There is no explanation of what became of the missing cases. This is a very high rate of missing data, and in such cases some way of handling this should be used, such as imputation, or a more complete discussion of the missing data and what bias it might yield should be included.

2. I have some concerns about the measure of liberalism of sexual attitudes scale. These seem to primarily to refer to views on premarital sex, but little is said about the relationship to marital sexual attitudes. As the DAS has items related to sexual satisfaction in the relationship, perhaps some additional analysis of how this relates to sexual attitudes scale would be useful.

3. Correlations of your measures with some respondent characteristics would help put the findings in context. What are the differences reported by males and females? Perhaps some differences by age and length of the relationship should also be reported here.

4. I can't see in the material I received that you will be providing access to the data set, at least the individual cases and responses of the respondents on the variables you used in this paper. This would be necessary for other researchers to explore the findings more completely using multivariate models.

6. PLOS authors have the option to publish the peer review history of their article (what does this mean?). If published, this will include your full peer review and any attached files.

Reviewer #1: Yes: Dr. Donna Miller

Reviewer #2: Yes: David R. Johnson

---

## [Author Response · Author response to Decision Letter 0]

15 May 2020

We have made the update of the manuscript format (first page, abstract and level 1 heading).

We have added the Romanian copy of the questionnaire (the English version was already included). 

3. We note you have included a table to which you do not refer in the text of your manuscript. Please ensure that you refer to Tables 2 and 3 in your text; if accepted, production will need this reference to link the reader to the Tables.

We have corrected these aspects.

Comments to the Author

1. Is the manuscript technically sound, and do the data support the conclusions?

Reviewer #1: Yes

Reviewer #2: Yes

2. Has the statistical analysis been performed appropriately and rigorously?

Reviewer #1: Yes

Reviewer #2: No

3. Have the authors made all data underlying the findings in their manuscript fully available?

Reviewer #1: Yes

Reviewer #2: No

We will upload the database with English labels. 

4. Is the manuscript presented in an intelligible fashion and written in standard English?

Reviewer #1: Yes

Reviewer #2: Yes

5. Review Comments to the Author

Reviewer #1: Being from the US, I would have liked more information on marriage in general in Romania. This would have been beneficial to having a better comprehension of the findings in the study.

In response to Reviewer 1’s request for more information on marriage in Romania we have added a chart showing statistical trends. To give wider context this is supplemented by data from other European countries. A comment on the likely reasons for the patterns shown has also been made.

Reviewer #2: There are some things that should be done to make the data analysis clearer and more complete.

1. Sample size issues. Although there were 455 persons who responded to the survey as indicated in your methodology section, the Ns reported in the tables were substantially less. In Table 1 the highest N was 282, and in Table 3 it was only 83. There is no explanation of what became of the missing cases. This is a very high rate of missing data, and in such cases some way of handling this should be used, such as imputation, or a more complete discussion of the missing data and what bias it might yield should be included.

The sample size issues in the data analysis arise from the composite definition of the factors analysed and the rate of participant response to specific component items in the questionnaire. This is now explained in the text of the paper. 

The point about potential bias has been addressed in the additional paragraph that reads as follows:

This restriction of usefulness in our available dataset does imply some degree of limitation on the robustness of any correlations that we detected, and this consideration may be felt to apply even more to the DAS and complex factor correlations presented later. However, while maintaining a measure of caution, although we cannot claim this subset of the data fully represented our larger original sample, the surviving positive correlations remain statistically significant findings. It is arguable that, in the absence of any bias arising from the incompleteness of the responses, those relationships would have emerged more clearly than they did from a restricted sample. 

2. I have some concerns about the measure of liberalism of sexual attitudes scale. These seem to primarily to refer to views on premarital sex, but little is said about the relationship to marital sexual attitudes. 

Little has been said about the relationship between liberalism of sexual attitude and marital sexual attitude because, for reasons now more explicitly explained, this dimension was not explored in the questionnaire administered. 

As the DAS has items related to sexual satisfaction in the relationship, perhaps some additional analysis of how this relates to sexual attitudes scale would be useful.

In response some further results of statistical analysis have been presented (pag 12). In the sense that they are negative results (no significant differences found) we are not clear how much they really add, and they would not conventionally be reported. But if they are felt to give helpful clarification we are happy to include them.

3. Correlations of your measures with some respondent characteristics would help put the findings in context. What are the differences reported by males and females? Perhaps some differences by age and length of the relationship should also be reported here.

Thank you very much for this recommendation. We have included an analysis of these aspects on pp 12-13.

4. I can't see in the material I received that you will be providing access to the data set, at least the individual cases and responses of the respondents on the variables you used in this paper. This would be necessary for other researchers to explore the findings more completely using multivariate models.

We have prepared the database with English labels and we will upload this to the system for full free access.

---

## [Decision Letter · Decision Letter 1]

16 Jun 2020

PONE-D-20-02203R1

SOCIOLOGICAL DIMENSIONS OF MARITAL SATISFACTION IN ROMANIA

PLOS ONE

Dear Dr. Delia,

Thank you for submitting your manuscript to PLOS ONE. After careful consideration, we feel that it has merit but does not fully meet PLOS ONE’s publication criteria as it currently stands. Therefore, we invite you to submit a revised version of the manuscript that addresses the points raised during the review process.

The revised version should take into account all remaining comments.

We look forward to receiving your revised manuscript.

Kind regards,

Petri Böckerman

Academic Editor

PLOS ONE

Reviewers' comments:

Reviewer's Responses to Questions

**Comments to the Author**

1. If the authors have adequately addressed your comments raised in a previous round of review and you feel that this manuscript is now acceptable for publication, you may indicate that here to bypass the “Comments to the Author” section, enter your conflict of interest statement in the “Confidential to Editor” section, and submit your "Accept" recommendation.

Reviewer #2: (No Response)

2. Is the manuscript technically sound, and do the data support the conclusions?

Reviewer #2: Partly

3. Has the statistical analysis been performed appropriately and rigorously? 

Reviewer #2: No

4. Have the authors made all data underlying the findings in their manuscript fully available?

Reviewer #2: No

5. Is the manuscript presented in an intelligible fashion and written in standard English?

Reviewer #2: Yes

6. Review Comments to the Author

Reviewer #2: There are still two things you need to do to make this an acceptable manuscript.

1. You only partly dealt with the missing data problem. Although using a missing data strategy such as multiple imputation would be most preferable and conform to the standard way in the social sciences to handle the high level of missing data in your survey, an acceptable alternative is to add a information on what items are leading to the most missing data. Do all items have about the same level of missing data on the DAS, or do a few contribute the most? It would not take much space to discuss present this and may help the reader understand the limits of the analyses with small sample sizes due to casewise deletion int statistical analyses.

2. Making the data available in English as well is very helpful. The only problem I see is that the items determining the gender and age of the respondent are not included in your list. They should be include because they would be very important for researchers wanting to use your date, and it is fair that they be included as you did use there variables in some of your analyses.

7. PLOS authors have the option to publish the peer review history of their article (what does this mean?). If published, this will include your full peer review and any attached files.

Reviewer #2: Yes: David R. Johnson

---

## [Author Response · Author response to Decision Letter 1]

7 Jul 2020

1. You only partly dealt with the missing data problem. Although using a missing data strategy such as multiple imputation would be most preferable and conform to the standard way in the social sciences to handle the high level of missing data in your survey, an acceptable alternative is to add a information on what items are leading to the most missing data.

Indeed, the analysis of the missing data is a very useful source of additional information; thank you very much for the recommendation about multiple imputation. In our database, most of the questions with missing data relate to items that belong to the factual area (the length of previous relationships, some details of the couple’s life and so on), and any estimation of missing responses, or a weighting of the answers, could have skewed the final results due to the small size of the sub-sample. We have included in the article a short explanation about the source of the missing data for building the composite variables.

Do all items have about the same level of missing data on the DAS, or do a few contribute the most? It would not take much space to discuss present this and may help the reader understand the limits of the analyses with small sample sizes due to casewise deletion int statistical analyses.

Indeed, the DAS components have up to 82 missing item responses, mostly from the questionnaire returns of young people lacking significant dyadic experience, but also from people who had completed only the first part of the questionnaire. The two items in the DAS with the highest levels of missing responses were I30 Not showing love (119 missing) and I29 Being too tired for sex (108 missing). Both of these are very personal questions and they were also at the very end of the questionnaire. We have now included in the paper a short explanation of these features.

2. Making the data available in English as well is very helpful. The only problem I see is that the items determining the gender and age of the respondent are not included in your list. They should be include because they would be very important for researchers wanting to use your date, and it is fair that they be included as you did use there variables in some of your analyses.

This is true and it was an error; we have uploaded an extended updated version of the database (in the first version of the article the analysis did not include age and gender).

We have uploaded the figure files into the PACE system.

---

## [Decision Letter · Decision Letter 2]

6 Aug 2020

SOCIOLOGICAL DIMENSIONS OF MARITAL SATISFACTION IN ROMANIA

PONE-D-20-02203R2

Dear Dr. Delia,

We’re pleased to inform you that your manuscript has been judged scientifically suitable for publication and will be formally accepted for publication once it meets all outstanding technical requirements.

Kind regards,

Petri Böckerman

Academic Editor

PLOS ONE

Additional Editor Comments (optional):

Reviewers' comments:

Reviewer's Responses to Questions

**Comments to the Author**

1. If the authors have adequately addressed your comments raised in a previous round of review and you feel that this manuscript is now acceptable for publication, you may indicate that here to bypass the “Comments to the Author” section, enter your conflict of interest statement in the “Confidential to Editor” section, and submit your "Accept" recommendation.

Reviewer #2: All comments have been addressed

2. Is the manuscript technically sound, and do the data support the conclusions?

Reviewer #2: Yes

3. Has the statistical analysis been performed appropriately and rigorously? 

Reviewer #2: Yes

4. Have the authors made all data underlying the findings in their manuscript fully available?

Reviewer #2: Yes

5. Is the manuscript presented in an intelligible fashion and written in standard English?

Reviewer #2: Yes

6. Review Comments to the Author

Reviewer #2: (No Response)

7. PLOS authors have the option to publish the peer review history of their article (what does this mean?). If published, this will include your full peer review and any attached files.

Reviewer #2: **Yes: **David R. Johnson

---

## [Editor Report · Acceptance letter]

10 Aug 2020

PONE-D-20-02203R2 

Sociological dimensions of marital satisfaction in Romania 

Dear Dr. Nadolu:

I'm pleased to inform you that your manuscript has been deemed suitable for publication in PLOS ONE. Congratulations! Your manuscript is now with our production department. 

Kind regards, 

on behalf of

Professor Petri Böckerman 

Academic Editor

PLOS ONE